# MicroRNA Associations with Preterm Labor—A Systematic Review

**DOI:** 10.3390/ijms25073755

**Published:** 2024-03-28

**Authors:** Adrianna Kondracka, Aleksandra Stupak, Magda Rybak-Krzyszkowska, Bartosz Kondracki, Anna Oniszczuk, Anna Kwaśniewska

**Affiliations:** 1Department of Obstetrics and Pathology of Pregnancy, Medical University of Lublin, 20-081 Lublin, Poland; adriannakondracka@wp.pl (A.K.); haniakwasniewska@gmail.com (A.K.); 2Department of Obstetrics and Perinatology, The University Hospital in Krakow, 30-551 Krakow, Poland; magda@hi-gen.pl; 3Department of Cardiology, Medical University of Lublin, 20-081 Lublin, Poland; 4Department of Inorganic Chemistry, Medical University of Lublin, 20-081 Lublin, Poland; anna.oniszczuk@umlub.pl

**Keywords:** MicroRNA expression, pregnancy, preterm labor

## Abstract

This systematic review delves into the connections between microRNAs and preterm labor, with a focus on identifying diagnostic and prognostic markers for this crucial pregnancy complication. Covering studies disseminated from 2018 to 2023, the review integrates discoveries from diverse pregnancy-related scenarios, encompassing gestational diabetes, hypertensive disorders and pregnancy loss. Through meticulous search strategies and rigorous quality assessments, 47 relevant studies were incorporated. The synthesis highlights the transformative potential of microRNAs as valuable diagnostic tools, offering promising avenues for early intervention. Notably, specific miRNAs demonstrate robust predictive capabilities. In conclusion, this comprehensive analysis lays the foundation for subsequent research, intervention strategies and improved outcomes in the realm of preterm labor.

## 1. Introduction

Preterm birth (PTB) is a complex clinical syndrome occurring before 37 weeks of gestation, with a variety of contributing factors. ‘It is the leading cause of mortality in children under five years old worldwide, resulting in over 1 million deaths each year. Clinical diagnosis of labor onset, whether term or preterm, relies on the observation of uterine contractions and is preceded by gradual cervical remodeling over several weeks. MicroRNAs (miRNAs), as critical regulators of gene expression, play a pivotal role in the epigenetic mechanisms that underpin both normal and abnormal pregnancy processes, including the intricate pathways leading to preterm labor. miRNAs are small, non-coding, single-stranded molecules consisting of 19–25 nucleotides, playing a role in mRNA stability and transcription. They act as essential regulators of gene expression across nearly all eukaryotes, with approximately 80% of human genes thought to be influenced by miRNAs. Different tissues, cells and bodily fluids have distinctive miRNA expression profiles, changing pathological processes and various disease states. MiRNAs are released into the circulation, remaining stable and easily measurable, making them potential biomarkers in conditions such as malignancy, infection and inflammatory disorders’ [1]. Recent investigations reveal that unique miRNA expression profiles, whether analyzed in the cervix or cervical cells, show potential in identifying women at risk of preterm delivery. However, the practicality of collecting samples from the cervix for broad obstetric population screening remains a challenge.

In the field of reproductive health, preterm labor presents a significant challenge, posing adverse consequences for the well-being of both mothers and infants. The intricate molecular mechanisms orchestrating the onset of preterm labor are a central area of focus, with microRNAs (miRNAs) emerging as integral components in this complex milieu. This systematic review aims to explore the intricate associations between miRNAs and preterm labor, undertaking a thorough analysis of existing research to shed light on the regulatory roles of miRNAs in gestational processes. Functioning as small, non-coding RNA entities, miRNAs intricately regulate gene expression, and their dysregulation is increasingly recognized as a potential contributor to preterm labor. Through a meticulous examination of the current literature, this review seeks to reveal patterns, identify key miRNA players and cultivate a nuanced understanding of the molecular factors influencing the timing and progression of preterm labor.

## 2. Methods

### 2.1. Inclusion Criteria

The following systematic review encompasses research published between 2018 and 2023, ensuring the incorporation of the most recent research findings. The focus will be specifically on studies that explicitly investigate microRNA, particularly those examining its role in pregnancy and preterm labor. The inclusion of studies within this timeframe is intended to capture the most recent developments and insights into the relationship between miRNA and preterm labor, facilitating a comprehensive and up-to-date analysis.

### 2.2. Exclusion Criteria

To ensure the quality and relevance of the selected studies, specific exclusion criteria are established. The exclusion of studies published in languages other than English is enforced to uphold consistency in language comprehension and interpretation. Additionally, studies that do not align with the specified objectives of investigating microRNA associations with preterm labor will be disregarded. To focus on comprehensive research articles and provide a substantial basis for analysis, studies exclusively presented as abstracts, conference proceedings and books will be excluded. This decision aims to eliminate preliminary or incomplete findings and guarantee a robust synthesis of information derived exclusively from peer-reviewed, comprehensive studies.

### 2.3. Search Strategy

Specific search strings were developed by combining the various keywords. The search strings used for the study included ‘miRNA and pregnancy’, ‘miRNA and preterm labor’, ‘miRNA clusters and preterm labor’, ‘differential expression of miRNA clusters and preterm labor’, ‘miRNA clusters and predictors of preterm labor’, ‘miRNA-based therapy and preterm labor’, ‘miRNA-targeted interventions and gestational complications’. Medline/PubMed and Science Direct were the databases that were used for retrieving the research articles. 

### 2.4. Study Selection

The study selection involved a two-step strategy. Initially, relevant articles were identified using keywords. Essential information was extracted and titles and abstracts were scrutinized for eligibility and relevance. In the second step, full-text articles were independently evaluated. PubMed was used to access complete texts and two reviewers independently screened the articles. Discrepancies were resolved through discussion, ensuring agreement before including studies in this review.

### 2.5. Quality Assessment

Two reviewers rigorously examined chosen articles to meet predetermined inclusion criteria. Each article underwent detailed quality and source database analysis. Selected articles were indexed in PubMed. The Mixed Methods Assessment Tool (MMAT) [2] assessed paper quality for the systematic literature review, ensuring comprehensive evaluation of both qualitative and quantitative studies. The MMAT summarizing the various studies included is shown in Table 1.

### 2.6. Data Extraction

Data extraction for each study was independently conducted, and a consensus was reached collaboratively by the two reviewers. Extracted data that covered microRNA, pregnancy, and microRNA association with preterm labor conclusions in this systematic review were drawn from study outcomes. Extraction relied on primary references, with cross-referencing performed by checking cited references. Notably, the selected study already included a summary of these references.

### 2.7. Study Selection

Various search strings were designed and used for searches in both databases. A total of 4976 references from Pubmed and 10,399 from ScienceDirect were identified. The search strings gave a very high reference count in Science Direct, many of which were non-specific. The references from the first step were screened for the time frame of the study and duplication. A total of 6887 references were eliminated based on these observations. Thus, a total of 8489 references were considered for step III. Of the 8488 references, around 6053 were eliminated based on their eligibility criteria. A further 2436 were sought for retrieval, out of which 2389 were eliminated based on their non-alignment with the objectives of the study. Thus, a total of 47 references were selected for writing the SLR of the study (Figure 1).

## 3. Results

The systematic review of microRNA associations with preterm labor offers insights into the diagnostic and prognostic potential of specific microRNAs. Cook et al. conducted a prospective study with 53 pregnant women, using the nCounter miRNA assay to identify plasma miRNA biomarkers for predicting preterm birth (PTB) and cervical shortening. Nine miRNAs, including hsa-miR-150-5p, were differentially expressed (*p* < 0.001) and exhibited strong predictive abilities for PTB (AUC = 0.8725) and cervical shortening (AUC = 0.8514). Validation in a cohort of 131 women confirmed these values, indicating the potential of plasma miRNAs in the first trimester as early predictors for PTB and cervical shortening, enabling timely intervention [1]. In another study, Mavreli et al. stressed the necessity for large-scale studies to assess the clinical significance of circulating miRNAs in pregnancy-related diseases. They advocated incorporating various biomarkers, including genomic, transcriptomic, proteomic and biophysical markers, along with maternal characteristics, to enhance future healthcare for pregnant women [3].

Addo et al. conducted a review on miRNAs in the placenta, focusing on human pregnancy-related diseases and environmental toxicant exposure. They emphasized the potential of miRNAs as biomarkers for exposure and disease, presenting a comprehensive overview of miRNAs in placental health and their vulnerability to environmental influences [4]. However, Ali et al. conducted a review on the role of miRNAs in placental development and their distinct expression in the placenta and maternal circulation during pregnancy complications. They identified potential miRNA biomarkers for predicting or diagnosing pregnancy complications [5].

Aryan et al. discussed the altered expression of maternal circulating miRNAs during pregnancy, particularly in the context of cardiovascular complications. The review highlighted miRNAs’ potential as diagnostic and prognostic biomarkers for cardiac dysfunction during and after pregnancy, with an ongoing exploration of miRNA-mediated regulation in pregnancy-related cardiovascular complications [6]. Chavira-Suárez et al. conducted a study examining circulating microRNAs (c-miRNAs) during and after pregnancy. The findings revealed reduced expression in maternal/fetal compartments throughout gestation compared to non-pregnant profiles. The global c-miRNA expression exhibited a bias associated with fetal sex in the first trimester, and a distinctive signature correlated with fetal growth, underscoring their involvement in various pregnancy-related processes [24].

In a separate investigation, Chen-Yun et al. utilized cardiac-specific miR-125b-1 knockout mice, revealing diminished miR-125b expression. This resulted in a 60% perinatal death rate and subsequent cardiac hypertrophy. Transcriptome and proteome analyses indicated downregulation of proteins related to fatty acid metabolism, contributing to reduced ATP production. The study suggested that miR-125b deficiency is associated with heightened neonatal mortality and cardiac hypertrophy, potentially attributed to dysregulated fatty acid metabolism [44]. Chen et al. identified four downregulated serum miRNAs (miR-139-3p, miR-196a-5p, miR-518a-3p and miR-671-3p) in placenta accreta spectrum (PAS) disorder. Integrating these miRNAs with clinical parameters significantly improved diagnostic accuracy (AUC 0.91, specificity 0.92). Functional analysis implicated these miRNAs in angiogenesis, embryonic development, cell migration, adhesion and tumor-related pathways [45].

Deng et al. observed decreased peripheral blood miR-196b expression in ectopic pregnancy (EP) patients, followed by an increase post-treatment, positively correlating with HCG, progesterone and estradiol. Risk factor analysis identified miR-196b as an independent risk factor for EP. A combined ROC curve, integrating miR-196b and clinical parameters, demonstrated high diagnostic accuracy (AUC 0.899). Functional analysis implicated miR-196b in angiogenesis, embryonic development and cell migration [46].

In a systematic review and meta-analysis, Dinesen et al. unveiled miRNAs associated with gestational diabetes mellitus (GDM). Several miRNAs, including miR-29a, miR-330, miR-134, miR-16, miR-223 and miR-17, were significantly upregulated, while miR-132 and miR-155 were decreased in GDM patients. These miRNAs hold promise as potential biomarkers for early GDM detection and offer insights into GDM pathophysiology [7]. Elhag and Khodor emphasized the pivotal role of microRNAs (miRNAs) in gestational diabetes mellitus (GDM) pathogenesis. Aberrant miRNA expression in the placenta and maternal blood suggested their utility as diagnostic and prognostic biomarkers. Specific miRNAs were identified as modulators of key pathways in glucose homeostasis, insulin sensitivity and inflammation, presenting potential targets for GDM diagnosis and therapeutic intervention [8].

Foley et al. explored maternal plasma exosomes during pregnancy, unveiling alterations in miRNA profiles, particularly related to energetic and metabolic processes, in both early and late pregnancy. These findings significantly contribute to understanding the mechanisms supporting a healthy pregnancy [26]. Gródecka-Szwajkiewicz et al. investigated angiogenic factors and miRNAs in umbilical cord blood, distinguishing between preterm and term newborns. Deviated concentrations of both pro-angiogenic and angiostatic factors were identified, along with dysregulated miRNAs (Angio-MiRs), shedding light on the observed ‘anti-angiogenic state’ in preterm newborns [47].

In the scrutiny of miR-204 in hypertensive disorders complicating pregnancy (HDCP), He and Ding identified elevated miR-204 levels as a robust diagnostic marker for HDCP, correlating with inflammatory factors. Elevated miR-204 expression independently increased the risk of adverse pregnancy outcomes, suggesting its potential as a prognostic indicator [27]. Hosseini et al. delved into miRNA expression in early pregnancy loss (EPL), revealing dysregulated miRNAs in maternal plasma and villous tissue associated with critical processes such as cell migration, proliferation and angiogenesis [28].

Hu and Zhang explored miRNAs in preeclampsia and intrauterine growth restriction (IUGR), emphasizing the involvement of dysregulated miRNAs in uteroplacental vascular adaptation. These miRNAs target genes crucial for trophoblast invasion, contributing to the pathogenesis of these complications [10]. Illarionov et al. conducted a prospective study on miRNA profiles in plasma during the first trimester, revealing discernible differences in levels among women at high risk of preterm birth. The study identified 15 dysregulated miRNAs, highlighting the potential utility of early miRNA profiles in predicting the risk of preterm birth [29].

Jin et al. highlighted the role of miRNAs in compromised pregnancies, such as pre-eclampsia, fetal growth restriction and gestational diabetes mellitus, offering potential therapeutic avenues [11]. Juchnicka and Kuzmicki emphasized significant miRNAs in gestational diabetes mellitus, highlighting their relevance as biomarkers and therapeutic targets in understanding the epigenetic regulation of GDM [12].

Légaré et al. comprehensively analyzed the plasma micro transcriptome in the first trimester, identifying pregnancy-specific miRNAs targeting pathways related to lipid metabolism, placenta development and embryo development, contributing to maternal metabolic adaptation [31]. Li et al. explored the role of miRNAs in immune cells during pregnancy, specifically decidual NK cells, revealing abnormal miRNA expressions associated with recurrent pregnancy loss, providing insights into immune dysregulation [13]. Liang et al. emphasized the placenta’s critical role in successful pregnancy and intrauterine development, investigating miRNAs as key regulators of trophoblast cell function and their association with common signalling pathways in pregnancy disorders [14].

Liu et al. focused on exosomal microRNAs (ExomiRs) as potential cargo molecules influencing gene expression and acting as prognostic biomarkers for pregnancy-associated complications, identifying specific ExomiRs associated with gestational diabetes mellitus (GDM) and their roles in diabetes progression [15]. Manuck et al. explored the association between nitric oxide (NO) pathway-related mRNA and miRNA expression and preterm birth, identifying differentially expressed genes in women delivering preterm versus at term. The study suggested the potential of miRNA-mRNA pairs for predicting prematurity, highlighting the association between maternal blood NO pathway-related molecules and preterm birth [32].

Masete et al. explored the connection between maternal diabetes and pregnancy complications, specifically in pregestational type 1 and type 2 diabetes mellitus. The review emphasized miRNAs’ emerging role in pregnancy-related disorders as potential diagnostic and predictive biomarkers, stressing the need for comprehensive miRNA profiling in various maternal diabetes types [16]. Mavreli et al. conducted a case-control study to identify differentially expressed miRNAs in first-trimester maternal plasma, aiming to predict spontaneous preterm delivery (sPTD). Specific miRNAs like miR-23b-5p and miR-125a-3p were identified as potential early predictors for sPTD, offering a foundation for non-invasive predictive tests and timely interventions [33].

Menon et al. presented a longitudinal study detailing changes in exosomal miRNA concentrations in maternal plasma between term and preterm births. The study identified miRNA signatures linked to gestational age, revealing alterations in exosomal miRNA content as potential biomarkers for pregnancy progression [34]. Omeljaniuk et al. explored the role of miRNA molecules in miscarriage, emphasizing their potential as early minimally invasive diagnostic biomarkers. The review discussed miRNAs’ involvement in vital cellular processes crucial for pregnancy, emphasizing their predictive role in monitoring early pregnancy complications, particularly after the first miscarriage [17].

Paul et al. reviewed miRNAs in the clinical context of human embryo implantation, exploring their expression in implanted versus non-implanted blastocysts and in the endometrium during different menstrual cycle phases. The study suggested miRNAs’ potential as biomarkers for predicting embryo implantation success [18]. Ramos et al. investigated miRNA expression in small extracellular vesicles (sEV) from peripheral blood, comparing term and preterm pregnancies. The study identified differential expression of miR-612 and other miRNAs associated with cellular senescence in preterm pregnancies, highlighting miRNAs’ potential role in preterm labor and premature rupture of membranes [35].

Salma et al. discussed the rise in pregnancy-related cardiovascular diseases, specifically postpartum stroke, emphasizing miRNAs’ predictive role in complications like preeclampsia. The review underscored miRNAs as potential diagnostic markers for cardiovascular risk during pregnancy [19]. Sawangpanyangkura et al. explored miR-223 expression in gingival crevicular blood (GCB) of pregnant women with and without gestational diabetes mellitus (GDM) and periodontitis. The study revealed distinct miR-223 expression in GCB and peripheral blood, indicating its potential involvement in the link between GDM and periodontitis [48].

Shekibi et al. delved into the molecular regulation of endometrial receptivity by miRNAs, offering insights into human and mouse miRNAs, particularly those in the Wnt signalling family. The review highlighted their pivotal role in modulating endometrial receptivity and discussed miRNAs’ potential as biomarkers and therapeutic targets for improving implantation rates [20]. Soobryan et al. sought early detection markers for pre-eclampsia and gestational hypertension, examining pro- and antiangiogenic factors and corresponding miRNAs. Specific changes in angiogenic factors and miRNAs linked to different gestational hypertensive subtypes were identified, suggesting potential molecular and pathological profiles [36].

Subramanian et al. addressed new tools and approaches for extracellular vesicle (EV) research, emphasizing EVs’ role in environmental stress responses. The seminar discussed EVs’ potential as biomarkers and challenges in identifying cell-specific EVs related to environmental exposures, stressing the need for rigorous reporting and cross-disciplinary approaches [21]. Sun et al. investigated miR-29b in gestational diabetes mellitus (GDM), evaluating its expression in placenta tissues. The study identified miR-29b as a potential regulator, with lower expression in GDM patient placentas. The involvement of miR-29b in cell growth and migration, possibly through HIF3A regulation, was highlighted [37].

Tian et al. aimed to identify key circulating microRNAs associated with missed abortion (MA) and investigate their role in the MA process. The study unveiled differentially expressed miRNAs linked to MA, suggesting their potential as circulating biomarkers for predicting and understanding missed abortion mechanisms [38]. Tong and Kaitu’u-Lino observed that miRs 374a-5p and let-7d-5p, while not necessarily clinical biomarkers, exhibited early derangement by 12–14 weeks gestation in pregnancies leading to preterm birth. This highlights a sensitive first-trimester period, suggesting the origins of fetal growth restriction proposed over 20 years ago [22].

Vonkova et al. explored miR-210 as a potential diagnostic and prognostic marker in acute fetal hypoxia, confirming significant upregulation in fetal blood during labor. This emphasizes the need for further investigation into its clearance and potential clinical applications [39]. Winger et al. conducted a retrospective nested case-control study predicting spontaneous preterm birth in an African American population using first-trimester peripheral blood maternal immune cell microRNA. Their findings, with an area under the curve of 0.80, indicate the potential of microRNA as a predictive tool for preterm birth [40].

Wommack et al. examined microRNA clusters’ association with preterm birth, identifying relationships within genomic loci. Clusters c14mc and c19mc demonstrated distinct associations with the length of gestation and infant outcomes, providing insights into the coordinated role of miRNA clusters in preterm birth [41]. Yang et al. focused on the potential application of peripheral blood exosomes and circulating miRNAs as disease-specific biomarkers in pregnancy complications and abnormal fetal development disorders, highlighting the growing interest in liquid biopsy as a diagnostic tool [23].

Zhao et al. investigated the association between circulating let-7 family miRNAs and ovarian response to stimulation in in vitro fertilization. Their results suggest a potential non-invasive predictive marker for ovarian response [42]. Burris et al. addressed the challenge of predicting spontaneous preterm birth, identifying a subset of upregulated microRNAs associated with subsequent spontaneous preterm birth. This study emphasizes the potential of microRNA expression as a predictive biomarker, with implications for reducing infant morbidity and mortality [43].

Table 2 consolidates findings from various review articles exploring the role of microRNAs (miRNAs) in pregnancy, particularly in the context of preterm labor and related complications. The studies reviewed span from 2018 to 2023, with each addressing different facets of miRNA research, including their potential as non-invasive biomarkers for pregnancy-related conditions [3], their involvement in crucial biological processes such as trophoblast invasion and uteroplacental vascular adaptation [10] and the impact of environmental toxins on miRNA expression [4]. Furthermore, the reviews delve into miRNAs’ roles in cardiovascular complications during pregnancy [6], the development of predictive models for pregnancy complications using circulating miRNAs and exosomes [23] and the exploration of miRNA-mRNA networks in conditions like preeclampsia and intrauterine growth restriction [5]. These papers emphasize the complex involvement of miRNAs in pregnancy, pointing to the necessity for further research to fully understand their roles and harness their diagnostic and therapeutic potential. Building upon the insights gleaned from review articles, our further analysis delved into original research findings presented in Table 3.

Several studies have documented miRNAs with increased expression levels in preterm labor cases, suggesting their involvement in the physiological alterations leading to preterm labor. Notably, miRNAs such as hsa-miR-150-5p, hsa-miR-374a-5p and hsa-miR-191-5p have been identified across different research designs and populations as significantly upregulated in preterm labor [1,21,47].

Conversely, a significant number of miRNAs have been reported to be significantly downregulated in preterm labor, such as hsa-miR-23b-5p and hsa-miR-125a-3p [33]. This indicates that the reduction in certain miRNAs may also contribute to the initiation of preterm labor.

The expression levels of miRNAs demonstrate dynamic fluctuations across the different trimesters of pregnancy, with specific miRNAs upregulated in the first trimester and others not showing differential expression until later stages [29]. This highlights the complex regulation of miRNA expression throughout pregnancy and its potential impact on preterm labor. Certain miRNAs, like hsa-miR-374a-5p and hsa-miR-191-5p, have been confirmed in multiple studies, reinforcing the consistency and potential significance of these miRNAs in relation to preterm labor [1,21].

The alterations in miRNA profiles have been linked to signaling pathways crucial for pregnancy, such as TGF-beta, p53 and glucocorticoid receptor signaling [34]. This association with key physiological pathways further underscores the role of miRNAs in the regulation of pregnancy and the onset of preterm labor (Table 3).

## 4. Discussion

The systematic analysis of microRNA (miRNA) associations with preterm labor yields valuable insights into potential diagnostic and prognostic markers for this significant pregnancy complication. The findings of Cook et al. underscore the potential of miRNA biomarkers for early intervention in preterm labor management, offering the possibility of mitigating adverse outcomes [1]. The significance of this study is highlighted by its alignment with the perspectives of Mavreli et al., who stress the importance of comprehensive investigations integrating various biomarkers. The inclusion of miRNA biomarkers enhances the potential for a personalized approach to healthcare for pregnant women. The identification of specific miRNAs as predictive markers opens avenues for the development of non-invasive diagnostic tools, representing a transformative shift in prenatal care. The integration of miRNA research into this study holds promise for advancing healthcare practices, with the potential to reduce the incidence and severity of preterm labor-related complications [3].

miRNAs such as hsa-miR-150-5p and hsa-miR-210, which are often upregulated in preterm labor, appear to mediate responses that could be detrimental to pregnancy maintenance. The upregulation of hsa-miR-150-5p, implicated in immune system regulation, may disrupt the delicate balance of maternal-fetal tolerance necessary for a successful pregnancy, suggesting a pathophysiological mechanism where elevated levels contribute to inflammatory responses unfavorable to gestation. On the other hand, the increase in hsa-miR-210 levels, associated with hypoxic conditions, suggests a mechanism by which placental insufficiency and stress could be exacerbated, leading to unfavorable pregnancy outcomes.

Conversely, miRNAs like hsa-miR-519d and hsa-miR-let-7g showcase roles that are seemingly protective or compensatory. These miRNAs are integral to cellular stress responses and the development of the placenta, indicating that their regulation is essential for maintaining pregnancy. They may stabilize the intrauterine environment and foster healthy placental growth, underscoring their potential as therapeutic targets to mitigate the risk of preterm labor.

Exploring the placental perspective is essential for comprehending the complexities of microRNA (miRNA) associations with preterm labor. Research by Addo et al. and Ali et al. emphasizes the vital role of miRNAs in reflecting environmental exposures and pathological conditions, offering a valuable avenue for early detection and intervention in cases of preterm labor. This aspect is particularly crucial in addressing the multifaceted nature of pregnancy complications [4,5]. Moreover, Chen-Yun et al. and Chen et al. provide insights that not only enhance the diagnostic capabilities of miRNAs but also illuminate the underlying molecular mechanisms involved in conditions leading to preterm labor [44,45]. Findings by Chavira-Suárez et al. reveal associations with fetal sex and growth, offering a nuanced understanding of miRNA dynamics about both maternal and fetal factors [24]. This holistic approach contributes to the comprehensive characterization of miRNA signatures, enhancing the precision of predictive models for preterm labor. These placental-focused studies are pivotal in the ongoing discussion on miRNA associations with preterm labor. This knowledge is instrumental in developing targeted interventions, personalized treatment strategies and advancing maternal-fetal healthcare practices.

In the realm of ectopic pregnancy (EP), Deng et al. identified miR-196b as an independent risk factor, suggesting its potential as a diagnostic marker. The acknowledgment of miR-196b as a distinctive risk factor implies the potential development of targeted diagnostic tools and interventions for the early detection and management of ectopic pregnancies, addressing a crucial aspect of reproductive health [46]. Furthermore, the extensive meta-analysis conducted by Dinesen et al. on miRNAs associated with gestational diabetes mellitus (GDM) significantly enhances our research. The inclusion of these findings in our study strengthens the broader scope of miRNA associations with various pregnancy complications, contributing to a more comprehensive understanding of their diagnostic and prognostic implications [7].

The exploration by Elhag and Khodor into aberrant miRNA expression in the placenta and maternal blood in the context of gestational diabetes mellitus further underscores the relevance of miRNAs as diagnostic and prognostic indicators [8]. Moreover, Foley et al.’s investigation of maternal plasma exosomes and the identification of miRNA profiles related to energetic and metabolic processes adds a crucial dimension to our discussion [26]. The implications of these findings contribute to our comprehension of the systemic impact of miRNAs on maternal health, particularly with metabolic processes during pregnancy.

Research exploring the connection between microRNAs (miRNAs) and distinct pregnancy complications is pivotal for the contemporary landscape of maternal and neonatal health. He and Ding offer valuable insights by pinpointing elevated miR-204 as a diagnostic indicator for hypertensive disorders complicating pregnancy (HDCP) [27]. This identification holds particular significance due to the escalating incidence of hypertensive complications during pregnancy, posing substantial risks to both maternal and fetal well-being. Jin et al. emphasize the central role of miRNAs in compromised pregnancies, including pre-eclampsia, fetal growth restriction and gestational diabetes mellitus [11]. In the current landscape, where these complications significantly contribute to maternal morbidity and neonatal mortality, understanding the intricate involvement of miRNAs establishes a foundation for developing precise diagnostic tools and therapeutic interventions. 

The investigation by Hosseini et al. into miRNA expression in early pregnancy loss (EPL) is paramount in the context of reproductive health. By unveiling dysregulated miRNAs associated with critical processes such as cell migration, proliferation and angiogenesis, this study sheds light on the molecular mechanisms underlying pregnancy loss [28]. Omeljaniuk et al. contribute to the contemporary understanding by discussing the potential of miRNAs as early minimally invasive diagnostic biomarkers for miscarriage [17]. In an era where advancing reproductive medicine aims to enhance the precision and timeliness of diagnostics, the exploration of miRNAs as potential biomarkers aligns with the current imperative for non-invasive, early detection methods. This study’s findings hold promise for transforming the clinical approach to miscarriage diagnosis, offering a more sensitive and timely assessment that can significantly impact patient care.

The physiological hasmpact of miRNA dysregulation in preterm labor involves more than the binary concepts of upregulation and downregulation. The reduction in protective miRNAs likehasa-miR-23b-5p has hsa-miR-125a-3p could lead to unchecked inflammatory responses and tissue remodeling processes, propelling the gestational tissue towards conditions favorable for preterm labor. This nuanced understanding of miRNA activity highlights the complexity of gestational regulation and the potential for targeted interventions.

The in-depth analysis conducted by Légaré et al. assumes considerable significance in this study, delving extensively into the complex interplay between microRNAs (miRNAs) and maternal metabolic adaptation. This investigation sheds light on pregnancy-specific miRNAs that intricately target pathways essential to lipid metabolism, placenta development and embryo development [31]. Equally pivotal is the study by Liang et al., whose clarification forms a foundation for understanding the molecular intricacies of trophoblast function, offering potential insights into therapeutic interventions for complications associated with pregnancy [14].

Moreover, the research by Liu et al. and Manuck et al., spotlighting the emergence of exosomal miRNAs as potential prognostic biomarkers in pregnancy-associated complications, holds particular significance [15,32]. These findings signify a shift in our approach to prognostic markers, suggesting the potential use of exosomal miRNAs as non-invasive indicators of complications during pregnancy. Adding to the significance is the work by Mavreli et al., which identifies specific miRNAs in first-trimester maternal plasma as potential early predictors for spontaneous preterm delivery (sPTD) [33]. This discovery holds transformative potential, indicating the feasibility of non-invasive predictive tests in the early stages of pregnancy. 

The interaction between miRNAs and critical signaling pathways, such as TGF-beta and p53, elucidates the molecular mechanisms potentially contributing to preterm labor. The modulation of TGF-beta signaling by miRNAs can impact placental structure and function, affecting fetal development and maternal immune tolerance. This interaction suggests a regulatory mechanism whereby miRNAs influence pregnancy outcomes through their effects on key cellular processes and signaling pathways.

Similarly, the role of miRNAs in regulating the p53 signaling pathway—a key regulator of cellular stress responses—underscores the importance of miRNA-mediated gene regulation in maintaining placental health and function. Dysregulation of this pathway through aberrant miRNA expression could contribute to placental insufficiency and an adverse intrauterine environment, highlighting the need for further research into the specific miRNA-target interactions that influence these critical pathways.

MiRNAs are increasingly recognized for their regulatory capacities in gene expression, significantly impacting placental formation and function. Studies have shown dysregulated miRNA expression profiles in various pregnancy complications, particularly preeclampsia (PE), which suggests a functional involvement in placental insufficiency and the pathophysiological processes therein.

For example, aberrant expression of specific miRNAs has been implicated in the misregulation of trophoblast invasion, angiogenesis and inflammatory processes within the placenta, all of which are critical aspects of placental function and potential points of dysfunction in PE. The ability of certain miRNAs to modulate these key placental functions indicates that they could serve as both biomarkers for placental dysfunction and targets for therapeutic intervention.

Moreover, a potential therapeutic angle for miRNAs involves their modulation to correct the pathological placental processes associated with conditions like PE. The diagnostic capabilities of miRNAs have been enhanced by the identification of unique expression patterns associated with various placental diseases, which allows for the development of non-invasive diagnostic tools and the promise of targeted miRNA-based therapies.

While promising, the clinical application of miRNAs in intercepting placental dysfunction requires more in-depth investigation. The specific roles of miRNAs in the placenta, their systemic effects and the translation of these findings into clinical practice pose substantial challenges that need to be addressed. This includes the need for targeted delivery mechanisms to the placenta and the complexity of miRNA interactions within the intricate network of placental gene regulation.

In summary, the evidence suggests that miRNAs hold the potential to impact placental function significantly and could be harnessed in the future to intercept placental dysfunctions such as PE. Ongoing and future research should focus on expanding our understanding of miRNA-mediated mechanisms in placental health and disease and exploring the therapeutic potential of miRNAs in placental dysfunction [49].

The study by Romero et. Al. presents compelling evidence that an increased sFlt-1/PlGF ratio at 28–32 weeks of gestation is associated with a heightened risk of spontaneous preterm birth (sPTB) and maternal vascular malperfusion, a finding that resonates with the results we observed in our analysis.

Specifically, the referenced paper elucidates the mechanisms by which sFlt-1 and PlGF, as biomarkers, might reflect the complex interplay of factors leading to placental dysfunction. An abnormal increase in the sFlt-1/PlGF ratio has been linked with impaired angiogenesis and placental insufficiency, contributing to the pathogenesis of conditions like sPTB and maternal vascular malperfusion. Our findings extend this perspective by suggesting that the temporal changes in this biomarker ratio can be indicative of the progression of placental pathology, with clinical ramifications apparent as gestation advances [50].

This review encapsulates studies exploring miRNA associations in various pregnancy-related disorders, such as gestational diabetes mellitus, hypertensive disorders, pregnancy loss and complications like preterm birth. These findings collectively underscore the diagnostic and prognostic potential of miRNAs, providing a foundation for further research, early intervention strategies and improved maternal and neonatal outcomes. 

Understanding the role of miRNAs in preterm labor extends beyond identifying their expression patterns; it necessitates a deeper exploration of their functional impacts, both protective and harmful and their interaction with pivotal signaling pathways. The differential expression of miRNAs offers a window into the complex regulatory mechanisms at play, providing potential targets for therapeutic intervention and the development of predictive biomarkers for preterm labor. Future studies should aim to unravel the specific mechanisms by which miRNAs influence the signaling pathways critical for pregnancy maintenance, paving the way for innovative approaches to manage and prevent preterm labor.

### Clinical Implication

In light of the challenges posed by the time-consuming and expensive nature of miRNA analysis, our findings underscore the importance of prioritizing specific miRNAs for the prediction and routine screening of preterm labor. The consistent upregulation of miRNAs such as hsa-miR-150-5p, hsa-miR-374a-5p and hsa-miR-191-5p across different studies and populations marks them as particularly promising biomarkers for identifying the risk of preterm labor. These miRNAs have not only demonstrated significant alterations in expression levels in cases of preterm labor but have also been validated across various research designs, suggesting their robustness and potential utility in clinical settings.

Furthermore, the identification of miRNAs significantly downregulated in preterm labor, such as hsa-miR-23b-5p and hsa-miR-125a-3p, adds another dimension to the miRNA profile that could be instrumental in predicting preterm labor. The dynamic nature of miRNA expression throughout pregnancy, with certain miRNAs being upregulated in the first trimester and others showing changes at later stages, emphasizes the necessity of a targeted approach in the selection of miRNAs for screening purposes.

Given the significant role of miRNAs in regulating key physiological pathways involved in pregnancy, such as TGF-beta signaling, p53 and glucocorticoid receptor signaling, it is recommended that hsa-miR-374a-5p and hsa-miR-191-5p be considered as primary candidates for further development into routine screening tests for preterm labor. Their consistent upregulation in preterm labor cases, coupled with their association with crucial signaling pathways, positions them as viable biomarkers for early detection and intervention strategies.

Implementing screening based on these specific miRNAs could enhance our predictive capabilities and allow for a more cost-effective allocation of resources, ultimately contributing to improved outcomes for mothers and their babies at risk of preterm birth. Future research should focus on validating these miRNAs in larger, diverse populations and developing streamlined protocols for their detection and analysis in clinical settings.

## 5. Conclusions

This systematic review has identified significant changes associated with preterm labor, highlighting the upregulation of specific miRNAs like hsa-miR-150-5p and the downregulation of others such as hsa-miR-23b-5p, indicating their crucial roles in pregnancy complications. These alterations in miRNA profiles are dynamically regulated across pregnancy trimesters and have been linked to key signaling pathways essential for pregnancy, like TGF-beta and p53 signaling. The consistency of these findings across studies emphasizes the potential of miRNAs as biomarkers for preterm labor.

Looking ahead, future perspectives necessitate longitudinal studies to authenticate miRNA markers, explore therapeutic interventions grounded in miRNA regulation and implement non-invasive diagnostic tests for effective preterm labor management. These initiatives are poised to propel the field of maternal and neonatal healthcare forward, fostering a more nuanced understanding of miRNA contributions to pregnancy complications and enabling targeted interventions for improved outcomes.

## Figures and Tables

**Figure 1 ijms-25-03755-f001:**
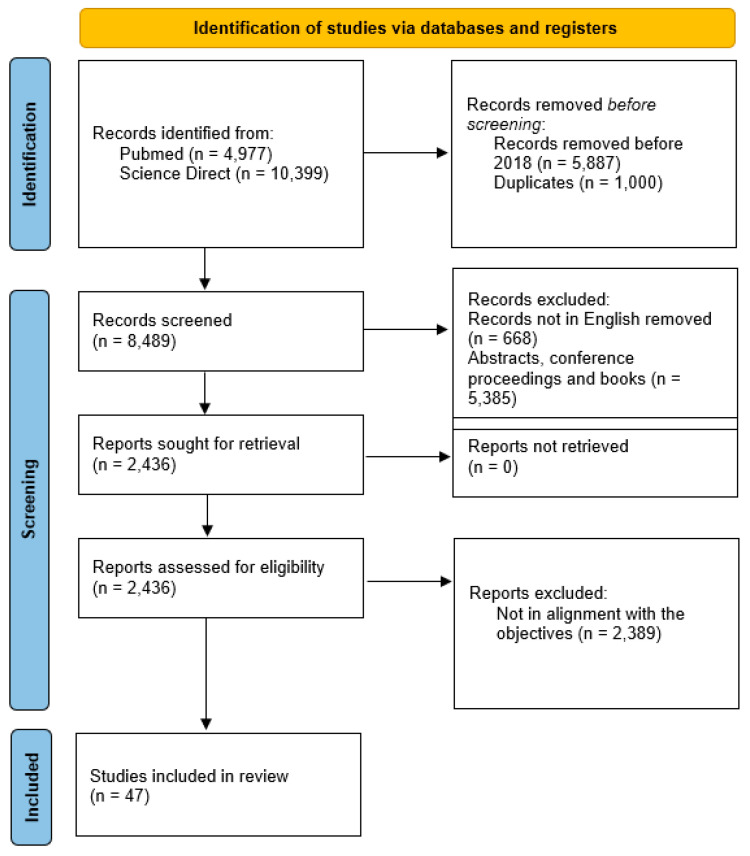
PRISMA flow-chart.

**Table 1 ijms-25-03755-t001:** Quality assessment of the study by the Mixed Methods Assessment Tool.

Category of Study Designs		Responses			
	Methodological Quality Criteria	Yes	No	Can’t Tell	
Screening questions (for all types)	S1. Are there clear research questions?	✔			
	S2. Do the collected data allow me to address the research questions?	✔			
	Further appraisal may not be feasible or appropriate when the answer is ‘No’ or ‘Can’t tell’ to one or both screening questions.				
1. Qualitative	1.1. Is the qualitative approach appropriate to answer the research question?	✔			[3,4,5,6,7,8,9,10,11,12,13,14,15,16,17,18,19,20,21,22,23]
	1.2. Are the qualitative data collection methods adequate to address the research question?	✔			
	1.3. Are the findings adequately derived from the data?	✔			
	1.4. Is the interpretation of results sufficiently substantiated by data?	✔			
	1.5. Is there coherence between qualitative data sources, collection, analysis and interpretation?	✔			
2. Quantitative randomized controlled trials	2.1. Is randomization appropriately performed?	✔			[1,24,25,26,27,28,29,30,31,32,33,34,35,36,37,38,39,40,41,42,43]
	2.2. Are the groups comparable at baseline?	✔			
	2.3. Are there complete outcome data?	✔			
	2.4. Are outcome assessors blinded to the intervention provided?	✔			
	2.5. Did the participants adhere to the assigned intervention?	✔			
3. Quantitative non-randomized	3.1. Are the participants representative of the target population?	✔			[44,45,46,47,48]
	3.2. Are measurements appropriate regarding both the outcome and intervention (or exposure)?	✔			
	3.3. Are there complete outcome data?	✔			
	3.4. Are the confounders accounted for in the design and analysis?				
	3.5. During the study period, is the intervention administered (or exposure occurred) as intended?	✔			
4. Quantitative descriptive	4.1. Is the sampling strategy relevant to address the research question?	✔			
	4.2. Is the sample representative of the target population?	✔			
	4.3. Are the measurements appropriate?	✔			
	4.4. Is the risk of nonresponse bias low?	✔			
	4.5. Is statistical analysis appropriate to answer the research question?	✔			
5. Mixed methods	5.1. Is there an adequate rationale for using a mixed methods design to address the research question?				
	5.2. Are the different components of the study effectively integrated to answer the research question?				
	5.3. Are the outputs of the integration of qualitative and quantitative components adequately interpreted?				
	5.4. Are divergences and inconsistencies between quantitative and qualitative results adequately addressed?				

**Table 2 ijms-25-03755-t002:** Overview of review articles included.

Authors & No of. Reference	Year of Publication	Aim	Conclusion
Mavreli et al., [3]	2018	Investigation of circulating miRNAs as promising noninvasive biomarkers for pregnancy-related complications.	Highlights the potential significance of circulating miRNAs as prognostic and diagnostic indicators for conditions associated with pregnancy.
Hu and Zhang, [10]	2019	An exploration of the impact of abnormal miRNA expression on genes related to trophoblast invasion and uteroplacental vascular adaptation in the context of preeclampsia and intrauterine growth restriction (IUGR).	Dysregulated miRNAs play crucial roles in compromising uteroplacental vascular function by targeting numerous genes involved in multiple signal transduction pathways.
Paul et al., [18]	2019	Recent progress in the exploration of miRNAs in human embryo implantation.	Additional investigation is necessary to explore novel investigative technologies for extracting minute quantities of miRNA and precisely identifying unidentified molecules.
Addo et al., [4]	2020	Summary of miRNAs investigated in the context of human pregnancy-related disorders and their association with exposure to environmental toxins in the placenta.	Altered expression of miRNAs in both the placenta and maternal plasma has been detected in connection with exposure to environmental chemicals. Their disruption is presumed to play a role in the onset of complications related to pregnancy.
Aryan et al., [6]	2020	Review recent literature on cardiovascular complications associated with pregnancy, particularly in initially healthy women. Explore the development of cardiac dysfunction and heart failure during and after pregnancy, emphasizing the involvement of microRNAs (miRNAs) in the underlying pathophysiological mechanisms.	Although the available data from various models of heart disease show promise, additional research is required to establish a direct and causal connection between miRNAs and cardiac pathophysiology in the context of cardiovascular complications during pregnancy. This deeper understanding will contribute to enhanced diagnostic methods and the creation of innovative therapeutic approaches.
Yang et al., [23]	2020	This study focuses on recent research regarding the roles of peripheral blood exosomes and circulating miRNAs in pregnancy complications and abnormal fetal developmental disorders. It particularly highlights the potential application of these exosomes and miRNAs as disease-specific biomarkers.	Advancements in medical science and technology enable the thorough collection of gestational data, aiding in the development of predictive models for pregnancy complications and fetal developmental disorders.
Ali et al., [5]	2021	A survey of uniquely expressed miRNAs in both the placenta and maternal circulation in cases of preeclampsia (PE) and intrauterine growth restriction (IUGR). These distinct miRNAs have the potential to function as biomarkers, offering promise in the anticipation and identification of complications during pregnancy.	By employing various bioinformatics tools, this investigation revealed potential target genes linked to both PE and IUGR through miRNAs. The study further elucidated the involvement of miRNA-mRNA networks in governing crucial signaling pathways and biological processes.
Jin et al., [11]	2022	Provide a summary of the existing knowledge regarding the expression patterns and regulatory functions of miRNAs in human placental development and associated diseases.	Identifying specific miRNAs at various gestational stages and within maternal circulation suggests the potential use of miRNAs as biomarkers for monitoring the progression of normal human pregnancy and detecting potential gestational diseases in clinical settings.
Juchnicka and Kuzmicki, [12]	2021	Examine the key miRNA molecules associated with gestational diabetes.	The alterations in miRNA expression in the blood during hyperglycemia-complicated pregnancy remain unclear, and additional investigations are necessary to establish a miRNA profile for predicting gestational diabetes mellitus (GDM).
Li et al., [13]	2021	Emphasized the role of miRNAs in overseeing the development and maturation of NK cells, as well as modulating the activation of NK cells and their subsequent activities, including the production of pro- or anti-inflammatory factors during pregnancy.	A more thorough comprehension of the regulatory mechanisms of miRNAs in decidual natural killer (dNK) and peripheral natural killer (pNK) cells, as well as their functional specificities, has the potential to establish a new framework for achieving a balance between immune function and damage at the maternal–fetal interface.
Liu et al., [15]	2021	Focused on summarizing the pathophysiological significance of miRNAs in the development of gestational diabetes mellitus (GDM) and their potential roles in clinical diagnosis and therapeutic approaches for GDM.	Additional research is required to validate miRNA profiles for the early prediction of gestational diabetes mellitus (GDM).
Masete et al., [16]	2022	Emphasizes the absence of miRNA profiling in pregnancies affected by T1DM and T2DM, underscoring the necessity for comprehensive miRNA profiling across various forms of maternal diabetes.	Subsequent investigations should prioritize miRNA profiling across various forms of maternal diabetes, aiming to elucidate the mechanisms underlying distinct types of diabetes during pregnancy.
Salma et al., [19]	2022	Comprehensive overview of existing knowledge and practices in this field, with a focus on physiological changes occurring during pregnancy.	The miRNA biomarker serves as a noninvasive, diagnostic and prognostic indicator for early detection of pregnancy-related cardiac disease and its complications.
Shekibi et al., [20]	2022	Summarized the key events of the endometrial cycle in humans and mice, followed by a review of the miRNAs identified to date in these two species, emphasizing their probable functional significance in the establishment of receptivity.	Subsequent investigations are necessary to explore the potential of miRNAs as biomarkers and/or therapeutic targets for the detection and enhancement of endometrial receptivity in human fertility treatment.
Elhag and Khodor, [8]	2023	Summarizes the existing information regarding the dynamics of microRNA (miRNA) during pregnancy, explores their involvement in gestational diabetes mellitus (GDM) and examines their potential applications as targets for diagnosis and therapy.	Altered expression of miRNAs in females with gestational diabetes mellitus (GDM) holds promise as noninvasive biomarkers, contributing to the discovery of fundamental mechanisms associated with gestational diabetes and related pregnancy complications.
Fu et al., [9]	2023	Examined the latest findings regarding the modulation of embryonic stem cells (ESCs), trophoblast cells and immune cells by microRNA, long non-coding RNA and circular RNA. Explored the potential utility of these non-coding RNAs as diagnostic and therapeutic indicators for pregnancy complications.	Further research should concentrate on exploring the functional aspects and molecular mechanisms of miRNAs, lncRNAs and circRNAs across various phases of pregnancy.
Liang et al., [14]	2023	Explores the involvement of miRNAs in modulating the function of trophoblast cells and various shared signaling pathways associated with miRNA regulation in pregnancy disorders.	Summarized microRNAs targeting respective mRNAs and governing the biological functions of trophoblasts in pregnancy disorders. These functions encompass trophoblast cell invasion, proliferation, migration, differentiation, apoptosis, autophagy, pyroptosis, ferroptosis, cellular metabolism and angiogenesis, operating through various pathways.
Omeljaniuk et al., [17]	2023	Summarize the current understanding of the involvement of miRNA molecules in the miscarriage process.	The assessment of potential miRNA molecules’ expression as minimally invasive diagnostic biomarkers can be conducted as early as the initial weeks of pregnancy.

**Table 3 ijms-25-03755-t003:** Overview of original articles included.

Reference	Study Design	Total Number of Participants	Methods	Conclusion	Results
Cook et al., 2019 [1]	Cohort study	511	Detect plasma miRNA biomarkers indicative of preterm birth and/or cervical shortening	The predictive efficacy of the miRNA biomarkers was validated in an independent cohort comprising 96 women with full-term deliveries, 14 with preterm deliveries, and 21 exhibiting early cervical shortening before 20 weeks of gestation.	Significantly higher mean relative expression: hsa-miR-150-5p, hsamiR-374a-5p, hsa-miR-19b-3p, hsa-miR-185-5p, hsa-miR-15b-5p, hsa-miR-191-5p, hsa-miR93-5p, hsa-let-7a-5p and hsa-miR-23a-3p at every time point except hsa-miR-191-5p, hsa-miR93-5p, hsa-let-7a-5p and hsa-miR-23a-3p at 12-14^+6^ weeks and hsa-miR93-5p at 15-18^+6^ weeks
Gródecka-Szwajkiewicz et al., 2021 [47]	Clinical trials	79	Ascertain if there are variations in the profiles of pro-angiogenic and anti-angiogenic factors in umbilical cord blood (UCB) between healthy preterm newborns appropriate for gestational age and term infants	The levels of five out of eight measured pro-angiogenic factors were notably reduced in umbilical cord blood (UCB) from preterm newborns. In contrast, two angiostatic factors showed significant upregulation in preterm UCB.	Healthy preterm newborns vs. term infants: 143 aberrantly expressed miRNAs: 33 miRNAs significantly upregulated (FC values ranged from 2.01 to 3.80) and 110 miRNAs significantly downregulated (FC values ranged from −2.01 to −6.20). Highest expression: has-miR-3135b Most downregulated: has-miR-941
Illarionov et al., 2022 [29]	Prospective study	24	miRNA profile in plasma during the initial and subsequent trimesters among pregnant women at a high risk of preterm birth	The miRNA profile in plasma during early pregnancy has the potential to forecast a heightened risk of preterm birth. This holds significance in the proactive prevention of gestational complications.	First trimester: Upregulated: hsa-miR-122-5p, hsa-miR-34a-5p, hsa-miR-34c-5p. Downregulated: hsa-miR-487b-3p, hsa-miR-493-3p, hsa-miR-432-5p, hsa-miR-323b-3p, hsa-miR-369-3p, hsa-miR-134-5p, hsa-miR-431-5p, hsa-miR-485-5p, hsa-miR-382-5p, hsamiR-369-5p, hsa-miR-485-3p, hsa-miR-127-3p) Second trimester: no differentially expressed miRNAs
Mavreli et al., 2022 [33]	Case-control study	68	Identify differentially expressed miRNAs in maternal plasma during the first trimester, aiming to pinpoint predictive biomarkers for spontaneous preterm delivery (sPTD) and enable timely interventions for this significant pregnancy complication	Statistical analysis showed that miR-125a is an independent early predictor for sPTL, offering a basis for developing a non-invasive test to assist clinicians in estimating patient-specific risk.	Downregulation: hsa-miR-23b-5p, hsa-miR-125a-3p
Menon et al., 2019 [34]	Cohort study	30	Detail the alterations in exosomal miRNA concentrations present in maternal plasma between mothers delivering term and preterm neonates throughout gestation, employing a longitudinal study design	The miRNA composition within circulating exosomes in maternal blood could potentially serve as a distinctive biomolecular fingerprint indicative of the progression of pregnancy.	A total of 173 miRNAs were found to significantly change across gestation for normal compared with PTB pregnancies. The differences in the miRNA profile targeted signaling pathways associated with TGF-b, p53 and glucocorticoid receptor signaling.
Ramos et al., 2023 [35]	Cohort study	31	Compare the expression of miRNAs in small extracellular vesicles (sEVs) from peripheral blood between term and preterm pregnancies	miRNAs derived from circulating sEVs exhibit distinct expression patterns in both term and preterm pregnancies. These miRNAs play a regulatory role in modulating genes within pathways that are pertinent to the pathogenesis of preterm labor (PTL) and preterm premature rupture of membranes (PPROM).	Higher expression: hsa-miR-612
Subramanian et al., 2023 [21]	Systematic review	555	Summarize literature, compile first-trimester circulating miRNAs tied to placental pregnancy complications and pinpointed top evidence-backed miRNAs as potential early biomarkers	Early detection and interventions in pregnancy complications can be facilitated through first-trimester biomarkers.	Higher expression: hsa-miR-374a-5p and hsa-miR-191-5p -replicated by 3 studies
Winger et al., 2020 [40]	Retrospective nested case-control study	486	Forecast spontaneous preterm birth in pregnant women within an African-American population by utilizing maternal immune cell microRNAs obtained in the first trimester from peripheral blood	Quantification of microRNA levels in peripheral blood during the first trimester enables the assessment of spontaneous preterm birth risk in both early and late samples of the first trimester of pregnancy within an African-American population.	12 microRNAs identify preterm birth risk in the first trimester: hsa-miR-181a-3p, hsa-miR-221-3p, hsa-miR-33a-5p, hsa-miR-6752-3p, hsa-miR-1244, hsa-miR-148a-3p, hsa-miR-1-3p, hsa-miR-1267, hsa-miR-223-5p, hsa-miR-199b-5p, hsa-miR-133b, hsa-miR-144-3p; hsa-miR-4485-5p, hsa-miR-340-5p, hsa-miR-132-3p, hsa-miR-219-5p also significant * no statement if up- or downregulated
Wommack et al., 2018 [41]	Population- based study	515	Explore how pregnancy-specific miRNA relates to gestation length and birth outcomes and Investigate if variations in coordinated circulating miRNA expression are linked to preterm birth (PTB)	Highlighting the necessity for additional research is crucial to elucidate the connections between circulating miRNAs and adverse pregnancy outcomes.	Positive correlation: hsa-miR-337-3p, hsa-miR-127-3p, hsa-miR-136-5p, hsa-miR-323a-3p, hsa-miR-543, hsa-miR-495-3p, hsa-miR-376c-3p, hsa-miR-376a-5p, hsa-miR-496
Burris et al., 2023 [43]	Prospective, nested, case-control study	74	Investigate the correlations between cervical microRNA expression and spontaneous preterm birth and to pinpoint a subset of microRNAs that can serve as predictive indicators for spontaneous preterm birth.	Researchers detected a rise in global microRNA expression and the upregulation of 95 unique microRNAs linked to subsequent spontaneous preterm birth.	There were 95 miRNAs associated with PTB: * all were upregulated * there were no significantly downregulated miRNAs 20 more highly expressed: hsa-miR-145, hsa-miR-1291, hsa-miR-30a-5p, hsa-miR-30d, hsa-miR-29a, hsa-miR-7, hsa-miR-942, hsa-miR-28-3p, hsa-miR-30e-3p, hsa-miR-23b, hsa-miR-542-3p, hsa-miR-29b, hsa-miR-21, hsa-miR-130b, hsa-miR-28, hsa-miR-425-5p, hsa-miR-181c, hsa-miR-9, hsa-miR-342-3p, hsa-miR-199b

## Data Availability

No new data were created or analyzed in this study. Data sharing does not apply to this article.

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
