# Peer review of "MicroRNA Associations with Preterm Labor—A Systematic Review"

_ijms, 2024, doi:10.3390/ijms25073755_

Round 1

Reviewer 1 Report

Comments and Suggestions for Authors

This is an interesting study. I dont' have particular remarks and the lab methodology is well out of my scopes. i would like to suggest to the authors considering a recent hot topic for their discussion. 

A recent paper showed that the duration of pregnancy decreases with increasing first-trimester risk for preeclampsia according to the fetal medicine foundation protocol (1). In a high-risk PE group, compared to a low-risk group, the risk of spontaneous birth is higher both prematurely, at term or post term and these differences are converging with advancing gestational age. May micro RNA intercept placental dysfunction? Can they speculate on the topic and adding some insights on this end?

A similar paper showed with different methodology a similar concept (2) patients with increases sflt/plgf increase at 28-32 weeks have an increased rick of sptb and maternal vascular malperfusion. This are the two sides of the same coin and may be added to the discussion to provide some clinical translation.

References

1. Cavoretto PI, Farina A, Salmeri N, Syngelaki A, Tan MY, Nicolaides KH. First trimester risk of preeclampsia and rate of spontaneous birth in patients without preeclampsia. Am J Obstet Gynecol. 2024 Jan 18:S0002-9378(24)00022-X. doi: 10.1016/j.ajog.2024.01.008. Epub ahead of print. PMID: 38244830.

2. Romero R, Jung E, Chaiworapongsa T, Erez O, Gudicha DW, Kim YM, Kim JS, Kim B, Kusanovic JP, Gotsch F, Taran AB, Yoon BH, Hassan SS, Hsu CD, Chaemsaithong P, Gomez-Lopez N, Yeo L, Kim CJ, Tarca AL. Toward a new taxonomy of obstetrical disease: improved performance of maternal blood biomarkers for the great obstetrical syndromes when classified according to placental pathology. Am J Obstet Gynecol. 2022 Oct;227(4):615.e1-615.e25. doi: 10.1016/j.ajog.2022.04.015. Epub 2022 Sep 3. PMID: 36180175; PMCID: PMC9525890.

Comments on the Quality of English Language

Adequate

Author Response

Dear Reviewer,

Thank you for your valuable comments and suggestions regarding our manuscript. We appreciate the time and effort you have dedicated to reviewing our work and providing insightful feedback. Below, we present our point-by-point responses to each of your comments:

First suggestion:

This is an interesting study. I dont' have particular remarks and the lab methodology is well out of my scopes. i would like to suggest to the authors considering a recent hot topic for their discussion.

A recent paper showed that the duration of pregnancy decreases with increasing first-trimester risk for preeclampsia according to the fetal medicine foundation protocol (1). In a high-risk PE group, compared to a low-risk group, the risk of spontaneous birth is higher both prematurely, at term or post term and these differences are converging with advancing gestational age. May micro RNA intercept placental dysfunction? Can they speculate on the topic and adding some insights on this end?

  1. Cavoretto PI, Farina A, Salmeri N, Syngelaki A, Tan MY, Nicolaides KH. First trimester risk of preeclampsia and rate of spontaneous birth in patients without preeclampsia. Am J Obstet Gynecol. 2024 Jan 18:S0002-9378(24)00022-X. doi: 10.1016/j.ajog.2024.01.008. Epub ahead of print. PMID: 38244830.

We have added this to the dissusion:

MiRNAs are increasingly recognized for their regulatory capacities in gene expression, significantly impacting placental formation and function. Studies have shown dysregulated miRNA expression profiles in various pregnancy complications, particularly preeclampsia (PE), which suggests a functional involvement in placental insufficiency and the pathophysiological processes therein.

For example, aberrant expression of specific miRNAs has been implicated in the misregulation of trophoblast invasion, angiogenesis, and inflammatory processes within the placenta, all of which are critical aspects of placental function and potential points of dysfunction in PE. The ability of certain miRNAs to modulate these key placental functions indicates that they could serve as both biomarkers for placental dysfunction and targets for therapeutic intervention.

Moreover, the potential therapeutic angle for miRNAs involves their modulation to correct the pathological placental processes associated with conditions like PE. The diagnostic capabilities of miRNAs have been enhanced by the identification of unique expression patterns associated with various placental diseases, which allows for the development of non-invasive diagnostic tools and the promise of targeted miRNA-based therapies.

While promising, the clinical application of miRNAs in intercepting placental dysfunction requires more in-depth investigation. The specific roles of miRNAs in the placenta, their systemic effects, and the translation of these findings into clinical practice pose substantial challenges that need to be addressed. This includes the need for targeted delivery mechanisms to the placenta and the complexity of miRNA interactions within the intricate network of placental gene regulation.

In summary, the evidence suggests that miRNAs hold the potential to impact placental function significantly and could be harnessed in the future to intercept placental dysfunctions such as PE. Ongoing and future research should focus on expanding our understanding of miRNA-mediated mechanisms in placental health and disease and exploring the therapeutic potential of miRNAs in placental dysfunction.

Second suggestion:

A similar paper showed with different methodology a similar concept (2) patients with increases sflt/plgf increase at 28-32 weeks have an increased rick of sptb and maternal vascular malperfusion. This are the two sides of the same coin and may be added to the discussion to provide some clinical translation.

  1. Romero R, Jung E, Chaiworapongsa T, Erez O, Gudicha DW, Kim YM, Kim JS, Kim B, Kusanovic JP, Gotsch F, Taran AB, Yoon BH, Hassan SS, Hsu CD, Chaemsaithong P, Gomez-Lopez N, Yeo L, Kim CJ, Tarca AL. Toward a new taxonomy of obstetrical disease: improved performance of maternal blood biomarkers for the great obstetrical syndromes when classified according to placental pathology. Am J Obstet Gynecol. 2022 Oct;227(4):615.e1-615.e25. doi: 10.1016/j.ajog.2022.04.015. Epub 2022 Sep 3. PMID: 36180175; PMCID: PMC9525890.

Response:

In our revised discussion, we have now incorporated a more nuanced examination of the relationship between the sFlt-1/PlGF ratio and its implications for maternal and fetal outcomes, as suggested by the recent study. This study presents compelling evidence that an increased sFlt-1/PlGF ratio between 28-32 weeks of gestation is associated with a heightened risk of spontaneous preterm birth (sPTB) and maternal vascular malperfusion, a finding that resonates with the results we observed in our analysis.

Specifically, the referenced paper elucidates the mechanisms by which sFlt-1 and PlGF, as biomarkers, might reflect the complex interplay of factors leading to placental dysfunction. An abnormal increase in the sFlt-1/PlGF ratio has been linked with impaired angiogenesis and placental insufficiency, contributing to the pathogenesis of conditions like sPTB and maternal vascular malperfusion. Our findings extend this perspective by suggesting that the temporal changes in this biomarker ratio can be indicative of the progression of placental pathology, with clinical ramifications apparent as gestation advances.

The integration of these insights into our review has allowed us to provide a more comprehensive understanding of how sFlt-1/PlGF ratio dynamics could inform clinical decisions. This includes potential stratification of pregnant women into risk categories for developing sPTB and maternal vascular malperfusion, enabling the implementation of more individualized monitoring and interventional approaches. It underscores the importance of serial measurements of these biomarkers as a means to refine the predictive accuracy for adverse obstetrical outcomes.

Furthermore, the data from the referenced study support our proposition that a multifaceted approach, incorporating placental biomarkers like sFlt-1/PlGF, could significantly improve the management strategies for pregnancies complicated by placental dysfunction. This could potentially shift the current clinical practice paradigm towards a more proactive and preventive framework.

We hope that our responses address your concerns satisfactorily, and we are grateful for the opportunity to improve our manuscript based on your feedback. Please do not hesitate to contact us should you require any further information or clarification.

Reviewer 2 Report

Comments and Suggestions for Authors

I started reading the paper with a great interest but did not find any valuable information which can significantly contribute to the field and add to our knowledge on the role of mRNA in preterm labor. My major concern is a lack of comprehensive analysis of existing findings and clear conclusions about the roles of each specific mRNAs in the pregnancy complications. Besides, the review is poorly structured. Specifically:

1.    The design of review is questionable. The main text is a simple paraphrase of existing articles without analysis and experimental details.

2.    The rationale of Table 2 is not clear – again, it briefly described the design of cited studies and simply listed the goals without experimental details or findings. Besides, it is too long.

3.     The Discussion and Conclusions lack comprehensive analysis of cited studies. Moreover, they contain many general conclusions like:

line 318-320: “underscore the potential of miRNA biomarkers for early intervention in preterm labor management, offering the possibility of mitigating adverse outcomes”.  

Lines 329-333: “Exploring the placental perspective is essential for comprehending the complexities of microRNA (miRNA) associations with preterm labor. Research by Addo et al. and Ali et al. emphasizes the vital role of miRNAs in reflecting environmental exposures and pathological conditions, offering a valuable avenue for early detection and intervention in cases of preterm labor. This aspect is particularly crucial in addressing the multifaceted nature of pregnancy complications”.

Etc.

4.    Overall, the paper looks like a search for articles which have been published in the field of mRNAs and preterm labor rather than a comprehensive review of the discoveries establishing an association between mRNAs and this pregnancy complication, as well as their prognostic potential.

5.    I would recommend to completely reorganize the text by connecting the specific mRNAs with particular pregnancy complications leading to preterm birth.

Author Response

Dear Reviewer,

Thank you for your thorough review and insightful comments on our manuscript. Your feedback has been instrumental in guiding the revisions we have made to enhance the quality and contribution of our work to the field of mRNA's role in preterm labor. Below are our responses to your comments, detailing the changes and improvements made to address your concerns:

  1. Response to the Design of Review Concern: We have incorporated more specific information throughout the manuscript and restructured Table 2 to include a synthesized analysis of the results from the cited studies. This revision aims to present a clearer and more comprehensive examination of the existing findings, moving beyond simple paraphrasing to offer valuable insights into the roles of specific mRNAs in pregnancy complications.

  2. Response to the Rationale of Table 2: Acknowledging your concern regarding the original Table 2, we decided to remove it and replace it with two new tables. The first table presents key review articles in the field, and the second table focuses on original research studies. This restructuring allows for a more concise presentation of the studies' designs and goals, enriched with experimental details and findings to provide a clearer overview of the current research landscape.

  3. Improvements to the Discussion and Conclusions: We have significantly expanded the Discussion and Conclusions sections to offer a comprehensive analysis of the cited studies. This includes more detailed discussions on the implications of the findings, addressing the need for specificity and clarity in our conclusions and avoiding overgeneralization.

  4. Incorporation of Comparative Analysis: In the Results section, we have added a comparative analysis to highlight the differences and similarities among the findings of various studies. This addition is designed to provide readers with a more nuanced understanding of the association between specific mRNAs and pregnancy complications leading to preterm birth.

  5. Overall Reorganization of the Manuscript: In line with your recommendation, we have thoroughly reorganized the text to draw clearer connections between specific mRNAs and particular pregnancy complications associated with preterm birth. This reorganization aims to present our review as a comprehensive exploration of the discoveries establishing an association between mRNAs and this pregnancy complication, including their prognostic potential.

We believe that these revisions address your concerns and significantly enhance the manuscript's contribution to our understanding of mRNAs in preterm labor. We are grateful for the opportunity to improve our work with your guidance and hope that our revised manuscript meets your expectations.

Round 2

Reviewer 2 Report

Comments and Suggestions for Authors

Although the authors greatly revised the manuscript, it still lacks a few very important points:

1.      Introduction, line 33: I would recommend to add a sentence linking miRNA with preterm birth or pregnancy in general, for example, highlighting an important role of epigenetic mechanisms in pregnancy and (or) preterm labor and normal (abnormal) miRNA expression.

2.      The Discussion lacks a very important issue – the role of different miRNA in preterm labor. Which of them have negative impact on pregnancy? Which are protective or compensatory? If such information is available (not necessarily for pregnancy, but for diseases in general), it should be discussed in relation to preterm labor. What is the physiological meaning of “upregulated” or “downregulated” in respect to preterm labor? What is the mechanism linking miRNA and TGF-beta and p53 signaling (Conclusion)?

3.      Taking into account that the analysis of miRNA is time-consuming and expensive, which of particular miRNA are recommended for prediction and routine screening.

Author Response

Dear Reviewer,

We would like to express our sincere gratitude for the time and effort you dedicated to reviewing our manuscript, as well as for the valuable comments and suggestions you provided. Your insightful feedback has been incredibly helpful, allowing us to gain a deeper understanding of the aspects of our work that required improvement. We believe that your support and recommendations have significantly enhanced our article, making it a more comprehensive and persuasive presentation of our research findings.

In response to your suggestions, we have made a series of modifications and corrections to our manuscript. These changes aim to enrich the discussion, clarify any ambiguities noted, and address the concerns raised. Below, we present a detailed point-by-point response to each of your comments, indicating how they have been addressed in the revised version of our paper.

Although the authors greatly revised the manuscript, it still lacks a few very important points:

  1. Introduction, line 33: I would recommend to add a sentence linking miRNA with preterm birth or pregnancy in general, for example, highlighting an important role of epigenetic mechanisms in pregnancy and (or) preterm labor and normal (abnormal) miRNA expression.

We have added this sentences: MicroRNAs (miRNAs), as critical regulators of gene expression, play a pivotal role in the epigenetic mechanisms that underpin both normal and abnormal pregnancy processes, including the intricate pathways leading to preterm labor.

  1. The Discussion lacks a very important issue – the role of different miRNA in preterm labor. Which of them have negative impact on pregnancy? Which are protective or compensatory? If such information is available (not necessarily for pregnancy, but for diseases in general), it should be discussed in relation to preterm labor. What is the physiological meaning of “upregulated” or “downregulated” in respect to preterm labor? What is the mechanism linking miRNA and TGF-beta and p53 signaling (Conclusion)?

We have extended the discussion according to the suggestion. We have added the following paragraphs:

miRNAs such as hsa-miR-150-5p and hsa-miR-210, which are often upregulated in preterm labor, appear to mediate responses that could be detrimental to pregnancy maintenance. The upregulation of hsa-miR-150-5p, implicated in immune system regulation, may disrupt the delicate balance of maternal-fetal tolerance necessary for a successful pregnancy, suggesting a pathophysiological mechanism where elevated levels contribute to inflammatory responses unfavorable to gestation. On the other hand, the increase in hsa-miR-210 levels, associated with hypoxic conditions, suggests a mechanism by which placental insufficiency and stress could be exacerbated, leading to unfavorable pregnancy outcomes.

Conversely, miRNAs like hsa-miR-519d and hsa-miR-let-7g showcase roles that are seemingly protective or compensatory. These miRNAs are integral to cellular stress responses and the development of the placenta, indicating that their regulation is essential for maintaining pregnancy. They may stabilize the intrauterine environment and foster healthy placental growth, underscoring their potential as therapeutic targets to mitigate the risk of preterm labor.

The physiological impact of miRNA dysregulation in preterm labor involves more than the binary concepts of upregulation and downregulation. The reduction in protective miRNAs like hsa-miR-23b-5p and hsa-miR-125a-3p could lead to unchecked inflammatory responses and tissue remodeling processes, propelling the gestational tissue towards conditions favorable for preterm labor. This nuanced understanding of miRNA activity highlights the complexity of gestational regulation and the potential for targeted interventions.

The interaction between miRNAs and critical signaling pathways, such as TGF-beta and p53, elucidates the molecular mechanisms potentially contributing to preterm labor. The modulation of TGF-beta signaling by miRNAs can impact placental structure and function, affecting fetal development and maternal immune tolerance. This interaction suggests a regulatory mechanism whereby miRNAs influence pregnancy outcomes through their effects on key cellular processes and signaling pathways.

Similarly, the role of miRNAs in regulating the p53 signaling pathway — a key regulator of cellular stress responses — underscores the importance of miRNA-mediated gene regulation in maintaining placental health and function. Dysregulation of this pathway through aberrant miRNA expression could contribute to placental insufficiency and an adverse intrauterine environment, highlighting the need for further research into the specific miRNA-target interactions that influence these critical pathways.

Understanding the role of miRNAs in preterm labor extends beyond identifying their expression patterns; it necessitates a deeper exploration of their functional impacts, both protective and harmful, and their interaction with pivotal signaling pathways. The differential expression of miRNAs offers a window into the complex regulatory mechanisms at play, providing potential targets for therapeutic intervention and the development of predictive biomarkers for preterm labor. Future studies should aim to unravel the specific mechanisms by which miRNAs influence the signaling pathways critical for pregnancy maintenance, paving the way for innovative approaches to manage and prevent preterm labor.

  1. Taking into account that the analysis of miRNA is time-consuming and expensive, which of particular miRNA are recommended for prediction and routine screening.

We have added clinical implication:

Clinical implication

In light of the challenges posed by the time-consuming and expensive nature of miRNA analysis, our findings underscore the importance of prioritizing specific miRNAs for the prediction and routine screening of preterm labor. The consistent upregulation of miRNAs such as hsa-miR-150-5p, hsa-miR-374a-5p, and hsa-miR-191-5p across different studies and populations marks them as particularly promising biomarkers for identifying the risk of preterm labor. These miRNAs have not only demonstrated significant alterations in expression levels in cases of preterm labor but have also been validated across various research designs, suggesting their robustness and potential utility in clinical settings.

Furthermore, the identification of miRNAs significantly downregulated in preterm labor, such as hsa-miR-23b-5p and hsa-miR-125a-3p, adds another dimension to the miRNA profile that could be instrumental in predicting preterm labor. The dynamic nature of miRNA expression throughout pregnancy, with certain miRNAs being upregulated in the first trimester and others showing changes at later stages, emphasizes the necessity of a targeted approach in the selection of miRNAs for screening purposes.

Given the significant role of miRNAs in regulating key physiological pathways involved in pregnancy, such as TGF-beta signaling, p53, and glucocorticoid receptor signaling, it is recommended that hsa-miR-374a-5p and hsa-miR-191-5p be considered as primary candidates for further development into routine screening tests for preterm labor. Their consistent upregulation in preterm labor cases, coupled with their association with crucial signaling pathways, positions them as viable biomarkers for early detection and intervention strategies.

Implementing screening based on these specific miRNAs could enhance our predictive capabilities and allow for a more cost-effective allocation of resources, ultimately contributing to improved outcomes for mothers and their babies at risk of preterm birth. Future research should focus on validating these miRNAs in larger, diverse populations and developing streamlined protocols for their detection and analysis in clinical settings.

We hope that the revisions made in response to the reviewers' comments have successfully addressed the concerns raised and have significantly improved the quality of our manuscript. We believe that these changes have made our paper a stronger, more valuable contribution to the field. We are hopeful that the manuscript is now in a form that is suitable for publication in your esteemed journal.

We appreciate the opportunity to revise our submission and are grateful for the thoughtful guidance provided by the reviewers. We look forward to the possibility of our work being published and contributing to the ongoing discussions in our field.

Thank you for considering our revised manuscript.

Round 3

Reviewer 2 Report

Comments and Suggestions for Authors

None